# Integrated people-centered eye care: A scoping review on engaging communities in eye care in low- and middle-income settings

Ling Lee [1,2,3☯] *, Elise Moo[1], Tiffany Angelopoulos[1], Aryati Yashadhana[1,4,5☯]

**1** International Programs Division, The Fred Hollows Foundation Australia, Melbourne, Australia, **2** School of Optometry and Vision Science, The University of New South Wales, Sydney Australia, **3** Department of Pediatrics, The University of Melbourne, Melbourne, Australia, **4** Centre for Primary Health Care and Equity, The University of New South Wales, Sydney, Australia, **5** School of Population Health, The University of New South Wales, Sydney, Australia

☯ These authors contributed equally to this work.
* Ling.lee1@unsw.edu.au

## Abstract

### Background

Community engagement has been endorsed as a key strategy to achieving integrated people-centered eye care that enables people and communities to receive a full spectrum of eye care across their life-course. Understanding the ways communities are engaged in eye care, to what degree participation is achieved, and the factors associated with intervention implementation is currently limited.

### Objective

The scoping review aimed to assess how community engagement is approached and implemented in eye care interventions in low- and middle-income countries, and to identify the barriers and facilitators associated with intervention implementation.

### Methods

Searches were conducted across five databases for peer-reviewed research on eye care interventions engaging communities published in the last ten years (January 2011 to September 2021). Studies were screened, reviewed and appraised according to Cochrane Rapid Reviews methodology. A hybrid deductive-inductive iterative analysis approach was used.

### Results

Of 4315 potential studies screened, 73 were included in the review. Studies were conducted across 28 countries and 55 targeted populations across more than one life-course stage. A variety of community actors were engaged in implementation, in four main domains of eye care: health promotion and education; drug and supplement distribution and immunization campaigns; surveillance, screening and detection activities; and referral and pathway

**Data Availability Statement:** All relevant data are within the paper and its Supporting Information files.

**Funding:** This work was supported by The Fred Hollows Foundation through the Advancing Integrated People-Centered Eye Care initiative, funded by the Australian Government's Australian NGO Cooperation Program. The funders had no role in study design, data collection and analysis, decision to publish, or preparation of the manuscript.

**Competing interests:** LL is currently a consultant to The Fred Hollows Foundation and has been a consultant to the International Agency for the Prevention of Blindness (outside the submitted work). EM is currently employed by the Fred Hollows Foundation and a postgraduate student at Monash University (outside the submitted work). AY has been a consultant to The Fred Hollows Foundation (within the submitted work). TA has no conflicts of interest to declare. There are no patents, products in development or marketed products to declare. This does not alter our adherence to PLOS ONE policies on sharing data and materials.

navigation. With the approaches and level of participation, the majority of studies were community-based and at best, involved communities, respectively. Involving community actors alone does not guarantee community trust and therefore can impact eye care uptake. Community actors can be integrated into eye care programs, although with varying success. Using volunteers highlighted sustainability issues with maintaining motivation and involvement when resources are limited.

## Conclusion

This scoping review provides researchers and policy makers contextual evidence on the breadth of eye care interventions and the factors to be considered when engaging and empowering communities in integrated people-centered eye care programs.

## Introduction

Eye health has been recognized as essential to achieving universal health coverage, and key to progressing a range of Sustainable Development Goals [1,2]. Vision impairment and blindness at any stage of life can significantly impact education outcomes, employment, health and well-being [3–5]. In 2020, the global prevalence of distance and near vision impairment (including blindness) was estimated to be 596 million and 510 million, respectively, and predicted to significantly increase due to the global pressures of ageing and global population growth [6]. The distribution and impact of this growth will be unequally borne by already disadvantaged and marginalized groups, including women and girls, who are more likely to experience blindness and vision impairment in their lifetime due to entrenched barriers to accessing services [1,6]. This places pressure on eye care services to reform current approaches, including the design and delivery of effective interventions, to meet the increasing demand, particularly in low- and middle-income countries (LMICs) where resources are restricted.

In 2019, the World Health Organization's (WHO) World Report on Vision recommended implementation of 'Integrated people-centered eye care (IPEC)', a continuum of health interventions, where services deviate from disease-oriented, siloed, vertical eye care programs, to holistic approaches that center people and promote a full spectrum of care across the life course [7]. One of the four strategies central to achieving the IPEC vision is empowering and engaging people and communities. Engaging communities and enabling eye health actors to work together on eye health issues has the potential to enhance health-seeking behaviors and practices, and contribute towards transforming the policies and environments that shape and determine eye health outcomes. Previous systematic reviews have explored various aspects of community engagement including ophthalmology participatory research [8], access to school-based eye care [9], and uptake of cataract surgery [10,11]. However, understanding the ways in which communities are engaged in eye care and to what degree community participation is achieved, is currently limited [1].

Shifting perspectives in eye care practice to enable holistic and intersectional approaches across the life course are key principles of achieving IPEC. Understanding public health issues from a life course perspective often involves longitudinal methods which can be time and resource intensive, resulting in a larger proportion of the literature being disease-oriented and confined to finite population groups. The scoping review presented in this article, aims to assess how community engagement is approached and implemented in eye care interventions conducted in LMICs, and where these interventions are applicable across different life course

stages. The review does not aim to measure the effectiveness of interventions reported in the studies reviewed. Rather, the review aims to identify the barriers and facilitators associated with intervention implementation across a range of diverse and complex contexts to understand what works for whom and under what circumstances, and its alignment with IPEC.

## Materials and methods

### Search strategy

The search strategy followed the Preferred Reporting Items for Systematic Reviews and Meta-Analyses for Scoping Reviews (PRISMA-ScR) and Cochrane Rapid Reviews guidelines [12,13], and was registered in OSF Registries (https://osf.io/tnwvj). The databases, Medline (OVID), Embase (OVID), Web of Science, Scopus and Cochrane Library, were searched for studies published between January 2011 and September 2021. A set of search terms (S1 Table) used for each area of interest were compiled. The database search results were imported into a single library in EndNote (Clarivate Analytics, USA) where duplicates were removed. The combined library was imported into Covidence systematic review software (Veritas Health Information, Australia) for title/abstract and full text screening.

For the search strategy and study selection, we drew from the WHO's Community Engagement Framework definition of community engagement which is described as a process of developing relationships that enable health system actors and community members to work together to address health-related issues and promote well-being to achieve positive health impact and outcomes [14]. We also used the framework to determine the different approaches to community engagement used within the included studies.

### Study selection

Studies were included in the review if they reported peer-reviewed empirical research or systematic reviews of interventions focused on engaging communities with either eye health promotion or care in LMICs. The World Bank's list of LMICs was applied [15]. Studies were excluded if: 1) no outcomes were reported; 2) articles were not published in the English language; 3) published over 10 years ago; 4) no full-text was available; or 5) conference abstracts.

Using the inclusion and exclusion criteria, titles and abstracts of all studies retrieved were dual-screened for 20% of the sample, and the remaining single-screened with a second reviewer scanning through a minimum 20% of excluded abstracts to address risk of selection bias (LL, AY, TA). Where it was unclear whether the selection criteria were met, studies were included for full-text review. As no systemic reviews passed the inclusion criteria, two reviewers independently reviewed five full-texts followed by a single independent reviewer for the remaining studies (LL, AY, TA). Any conflicts during screening and full-text review were resolved by a third reviewer.

### Quality appraisal

The QualSyst tool [16] was used to appraise the quality of mixed-methods empirical research included for data extraction. Quality appraisal was conducted by one reviewer and verified by a second reviewer (LL,TA). Risk of bias across studies was not conducted for the scoping review, due to the lack of homogeneity of reported outcomes.

### Data extraction, analysis & synthesis

The categorical data extracted included country of study, study design, targeted condition(s), targeted life course stage(s) according to study population. Papers were also categorized using

**Table 1. Community engagement 'approaches' and 'levels' used in deductive analysis.**

| Community Engagement Approaches | |
|---|---|
| Level 1 – Oriented | The community is informed and mobilized to participate in addressing immediate short-term concerns with strong external support. |
| Level 2 –Based | The community is consulted and involved to improve access to health services and programs by locating interventions inside the community with some external support. |
| Level 3 – Managed | There is collaboration with leaders of the community to enable priority settings and decisions from the people themselves with or without external support of partners. |
| Level 4 –Owned | Community assets are fully mobilized, and the community is empowered to develop systems for self-governance, establish and set priorities, implement interventions and develop sustainable mechanisms for health promotion with partners and external support groups as part of a network. |
| **Community Engagement Levels of Participation** | |
| Inform | Community members are informed about the intervention but are not involved in the design or implementation. Primarily delivered by external organizations or actors. |
| Consult | Community members are consulted to inform the design and/or implementation of the intervention by external organizations or actors. |
| Involve | Community members are involved in planning and/or implementation, with strong external direction and support. |
| Collaborate | Community members collaborate with external organizations or actors to design, plan, and implement the intervention, with a focus on leveraging local skills and knowledge. |
| Empower | Community members act collectively in all aspects of implementation with the broader aim to gain greater control over their health and the quality of life. Control and governance of interventions are localized and seek to embed existing leadership structures. |

the community engagement approaches, and five levels of community participation, outlined in the WHO Community Engagement Framework [14]; both of which were conducted by a single reviewer and verified by a second reviewer (Table 1). Descriptive statistics were used to analyze categorical data.

Inductive analysis was performed on each included study in the final synthesis, whereby any references (qualitative or quantitative) to community engagement interventions or intervention components and their outcomes, were coded using NVivo qualitative data analysis software (QSR International Pty Ltd. Version 12, 2018). An iterative approach to inductive coding was applied to capture detail on the types of community engagement activities that were applied within interventions, and to identify contextual factors impacting implementation. Where applicable, factors were also coded as a 'barrier' or 'facilitator' (to implementation).

## Results

### Overview and descriptive results

The database search identified 4315 potential studies. After removal of duplicates 2651 titles and abstracts were screened. Of these, 269 full-text publications were retrieved for consideration. A total of 196 articles were excluded after full-text review, leaving 73 studies for inclusion. A PRISMA flowchart is presented in Fig 1. The characteristics of the included studies are outlined in S2 Table.

Studies were conducted across 28 different countries, with 20.5% (15/73) from India. Interventions involved improving vision impairment/blindness prevention or promotion, or by targeting specific eye conditions, and neglected tropical diseases or infectious diseases that can affect the eyes. Onchocerciasis was the most targeted condition (18/73), particularly in Sub-Saharan Africa (10 countries).

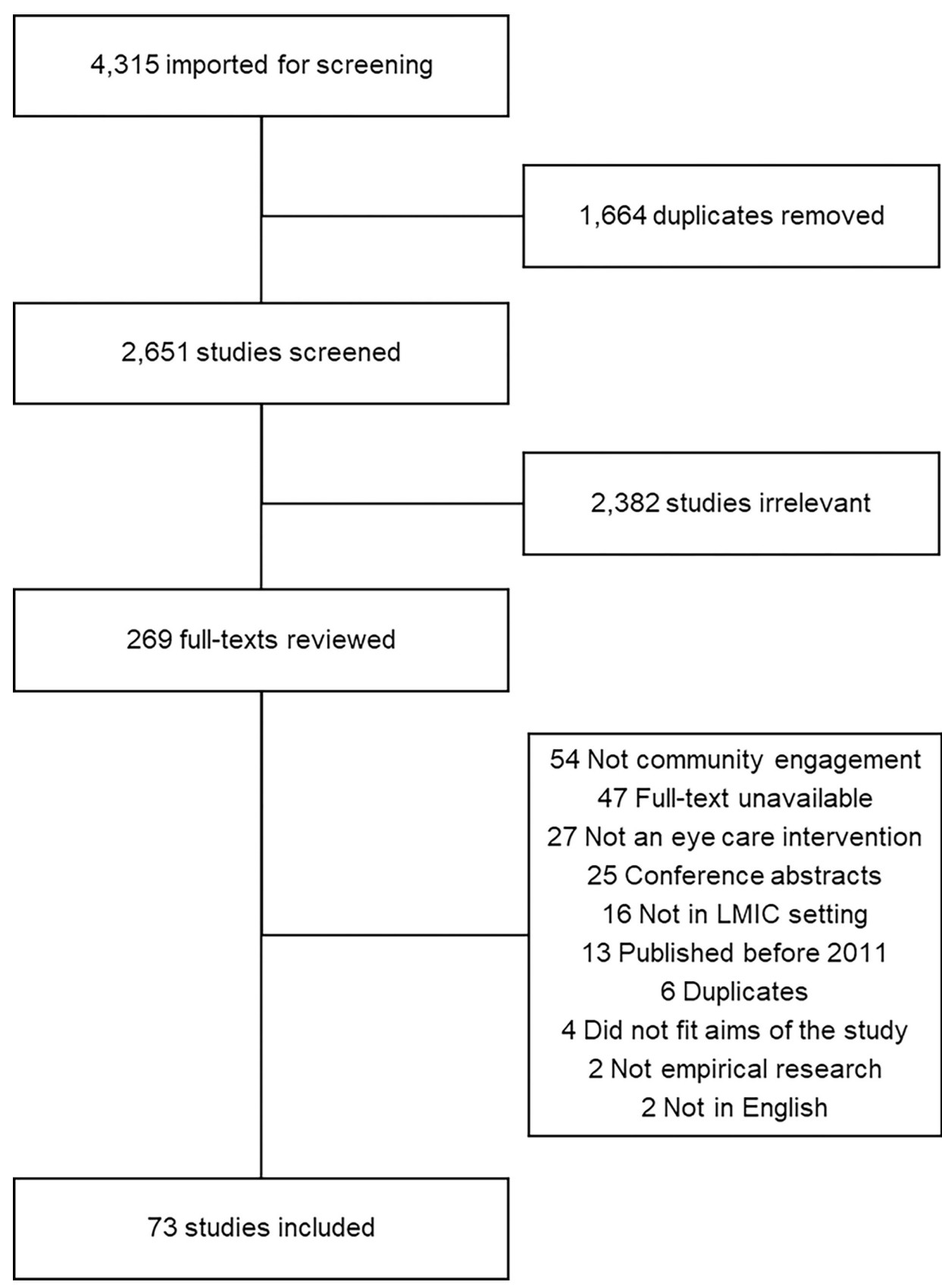

**Fig 1. PRISMA flow chart of review process and sampling.**

**Table 2. Community engagement activities and the level of participation (N = 73).**

| Activities engaging communities | Inform | Consult | Involve | Collaborate | Empower | Total |
|---|---|---|---|---|---|---|
| Health promotion and education | 4 | 0 | 9 | 4 | 0 | 13 |
| Drug and supplement distribution and immunization campaigns | 0 | 0 | 4 | 1 | 2 | 7 |
| Surveillance, screening and detection activities | 0 | 0 | 7 | 1 | 0 | 8 |
| Parental referral and pathway navigation | 0 | 0 | 1 | 0 | 0 | 1 |
| Intervention buy-in and involvement | 0 | 1 | 1 | 1 | 0 | 3 |
| Multiple activities | 1 | 2 | 23 | 7 | 4 | 37 |
| **Total** | 5 | 3 | 45 | 14 | 6 | 73 |

Fifty-five studies used a community engagement approach where the study population spanned across more than one life course stage. Interventions most frequently targeted the childhood stage, with very few targeting before birth or pre-natal care.

The studies revealed four distinct eye health intervention domains where communities were engaged in implementation including:

- Health promotion and education

- Drug and supplement distribution and immunization campaigns,

- Surveillance, screening and detection activities

- Referral and pathway navigation

Approximately half (37/73) of the articles engaged communities in more than one activity. In the sections to follow we present, for each of these intervention domains, the community engagement approaches identified, and main implementation barriers and facilitators reported. With the interventions' community engagement approaches, four were community-oriented, 44 community-based, 21 community-managed and four community-owned. For the level of participation, the majority of studies (45/73), at most, involved communities. Six studies empowered communities (Table 2), four of which were onchocerciasis community-directed treatment interventions, and two rehabilitation projects.

## Health promotion and education

More than half of the reviewed articles (n = 42) used community engagement approaches to deliver interventions focused on health promotion and education, or included a health promotion and education component within a multidimensional intervention. Activities included advocacy campaigns, strengthening health infrastructure to promote facial cleanliness, providing additional health education materials, and using non-health workers and community health workers (Table 3).

**Factors related to engagement of community in implementation.** Restricted time and capacity of different community-based actors to deliver interventions was identified as a barrier, including teachers providing school-based eye health programs [19,20], and local peer supporters providing a peer-to-peer diabetic retinopathy education program [46]. Designing interventions to be responsive to community contexts and needs were identified as key to implementation success, including recruiting local volunteers with in-depth knowledge of their communities in school-based eye health programs [23], diabetic retinopathy peer support groups [45,46], and utilizing local media for promotion [36,46,47]. The ability for intervention implementation to be flexible and make rapid adjustments based on local guidance facilitated community engagement [57].

Table 3. How community actors engaged in eye health promotion and education.

| Community actor(s) | Health promotion & education role(s) |
|---|---|
| Teachers | • Deliver eye health education in schools [17–20]<br>• Promote/supervise outdoor activities for myopia progression prevention [21,22]<br>• Be vision ambassadors [23]<br>• Train other teachers in vision screening [24]<br>• Promote hygiene practices and train student hygiene ambassadors [25] |
| Volunteers | • Deliver or facilitating eye health/immunization education or health promotion in schools/community [26–35]<br>• Deliver peer support groups for people with diabetes<br>• Utilize local media for eye health promotion and advocacy [36–38]<br>• Mobilize community members for eye care, diabetes care and immunization uptake [31,39–43]<br>• Discuss trachoma trichiasis care with community members [44] |
| Peers | • Deliver peer support groups for people with diabetes [45,46]<br>• Utilizing local media for eye health promotion [47]<br>• Delivery hygiene and sanitation education to peers [48]<br>• Apply peer pressure to carers to immunize their children [49]<br>• Be vision or trachoma ambassadors [23,25,50] |
| Community health workers | • Provide diabetic retinopathy sensitization/education [51,52]<br>• Provide activities and education for carers to engage with vision impaired children [53]<br>• Displaying health promotion material [54]<br>• Deliver eye health education in community [50,55] |
| Community and religious leaders | • Delivering eye health and vaccination education within the community [29,38,56]<br>• Involved in developing advocacy campaign and education materials [57] |
| Local artists | • Develop eye health education material [19]<br>• Provide eye health education through performance [47,58] |
| Patients | • Using multimedia to improve cataract surgery informed consent [59] |

Availability and training of community members to encourage trichiasis surgery uptake [29], and vision impairment programs [53] facilitated accessibility of interventions. A lack of information being provided by community-based volunteers to community members led to reduced uptake of trichiasis surgery [29]. Comprehensive training provided to peer supporters in a diabetic retinopathy program increased confidence and self-fulfillment, enabling self-efficacy and appropriate delivery of the intervention [46].

**Factors related to engagement of community members in the uptake of interventions.** Understanding community power structures and sociocultural norms were key to gaining community support of eye health promotion interventions [38], and facilitating their engagement in interventions such as spectacle compliance [55] and onchocerciasis treatment [32]. In Egypt, local government hospitals were perceived by community members as providers of lower quality of care, impacting a trichiasis surgery uptake [31]; while in Tanzania, women were more likely to report challenges with accessing trichiasis surgery [29]. Engaging and involving local community members in school eye health programs [23], and trichiasis surgery uptake through literacy level appropriate information [29], also led to greater trust and acceptability of interventions. Activities that supported motivation of community volunteers and teachers in intervention implementation included monitoring visits [19], free eye health education [23], and encouraging sustainability plans for hygiene education for trachoma prevention [25]. Experiencing or perceiving a tangible impact as a result of successful vision screenings [24] or health promotion interventions [46] were also identified as motivating factors for community volunteers.

In Kenya, a lack of financial remuneration for peer-supporters involved in diabetic retinopathy education and screening [46] was a barrier to implementation. This differed in India

where financial incentives provided to 'Accredited Social Health Activists' for community mobilization in a diabetic retinopathy intervention, and community treatment assistants in trichiasis detection in Tanzania [44] facilitated engagement.

Geographic access to hard-to-reach communities was identified as a key barrier to community engagement, including promotion of, and mobilizing participation in eye health camps in India [36]. Community volunteers delivering a support program for Malawian parents of children with vision impairment were only able to provide the intervention to participants within a one-hour travel time range [53]. Drought caused disruptions to the delivery of trachoma prevention programs in Kenya [25].

## Drug and supplement distribution and immunization campaigns

Twenty-five reviewed articles engaged communities in drug and supplement distribution, and immunization campaigns. Table 4 details the community actors and their role(s) in the distribution and campaign interventions.

**Factors related to engagement of community in implementation.**   Intervention timing was central to community engagement in drug distribution and immunization interventions. Time needed for the procurement, supply and distribution of ivermectin for onchocerciasis control in Nigeria [60], Democratic Republic of Congo [33], Cameroon [37,61,68,71], and Tanzania [32], and measles-rubella vaccination campaigns in rural Bangladesh [57], caused delays and complexities in implementation. Allowing ample time for community sensitization before the intervention occurred, including sensitizing parents to immunization campaigns [49,57]; promoting mass drug administration for onchocerciasis in central places such as markets or places of worship [60,66]; and involving community leaders or members in planning [33,37,66] was pivotal to intervention success or failure.

Designing interventions to be responsive to community contexts and needs was identified as a key factor in implementation. For example, not involving community leaders [33,37] or local implementing partners [27] in intervention planning limits legitimacy and capability to plan the timing of drug distribution and immunization campaigns effectively, subsequently restricting community engagement. Ivermectin administration interventions were criticized for being short-lived, lacking promotion of onchocerciasis prevention and control beyond the

**Table 4. How community actors engaged in drug and supplement distribution and immunization campaigns.**

| Community actor(s) | Drug and supplement distribution and immunization campaign role(s) |
|---|---|
| Volunteers | • Sensitize, mobilize and distribute antibiotics for onchocerciasis treatment to community members [32,60–64]<br>• Distribute antibiotics for trachoma treatment [44]<br>• Mobilize communities for vitamin A supplementation or immunization [57,65] |
| Peers/Community members | • Recruit/select community distributors for onchocerciasis treatment [32,33,37,62,66–68]<br>• Determine how to motivate community drug distributors [37]<br>• Assist health workers with vitamin A supplementation campaigns [69]<br>• Determine when or how onchocerciasis treatment occurs within the community [32,68] |
| Community and religious leaders | • Communicate onchocerciasis treatment approach<br>• Select community distributors for trachoma treatment [44] |
| Local guides | • Assist vaccination teams to locate nomadic and pastoralist communities [34] |
| School staff (principals and teachers) | • Promote and provide space for immunization campaigns [49] |
| Community health workers | • Sensitize, mobilize and distribute antibiotics for onchocerciasis treatment to community members [70,71]<br>• Sensitize and mobilize communities for immunization [54] |

life of the project [61], and ignoring pressing community issues such as poverty and poor infrastructure, leading to community disengagement [60].

Trust also impacted implementation. Distrust of ivermectin treatment within the Nigerian Ministry of Health created barriers for community volunteers in delivering mass drug administration campaigns [71], however enabling a sense of community ownership over the intervention allowed implementation nonetheless [64,71].

In multiple Sub-Saharan African countries, general human resource issues limited the availability of local and mid-level health workers to provide supervision or training in drug distribution [62,64,66,69], subsequently limiting the availability of community distributors of ivermectin [33,37,61]. Insufficient length and quality of training received by community volunteers was also identified as a key barrier leading to disengagement and high turnover [26,33]. Simple intervention protocols allowed those with lower literacy levels to engage as community volunteers [26,71]. The provision of literacy level-appropriate health information by community health workers also enabled vaccination uptake [40]. Sociocultural facilitators to community volunteer motivations included wanting to help their own communities [63,71], or receiving assistance with farm work [71]. From a gender perspective, men were less likely to attend training to be community mobilizers in immunization campaigns [27], and male community distributors of ivermectin were less likely to access female community members due to either the head of the household needing to be present but unavailable at the time, or women seeking alternative treatments [60,63].

A lack of incentives and resource support, including financial reimbursement for community volunteers' or distributors' time presented a key economic barrier to community engagement. This included community volunteers involved in immunization campaigns in Malawi [27], Somalia [34], and Bangladesh [57], and community distributors of onchocerciasis treatments in the Democratic Republic of Congo [33], and Cameroon [37,61]. Lack of remuneration or resources for community distributors led to low involvement [33], poor attrition, dissatisfaction [61], and recruitment [63,64]. As community distributors were often living in poverty, in some cases, donations for ivermectin treatment were requested which decreased uptake [60,63]. Conversely in India 'Anganwadi workers' in a measles immunization campaign [54], enabled more time to be dedicated to the intervention when remunerated, facilitating campaign success.

In Nigeria [60], Cameroon [61,66] and Somalia [34] difficult terrain and transport costs had restricted geographic access to rural 'hard-to-reach' communities. Immunization campaigns in Bangladesh [57] and Somalia [34], and mass drug administration for onchocerciasis in Nigeria [60] and the Democratic Republic of Congo [72] were disrupted or hampered by political unrest in certain areas.

**Factors related to engagement of community members in the uptake of interventions.** Similarly, once treatments had been acquired, failure to engage community members on appropriate timing of mass distribution limited intervention uptake [32,37,60,61,63,64,68]. In several studies community volunteers reported many potential participants were not home to receive treatment due to farming [32,63,64,68]. Other issues related to intervention timing included varying and multiple campaigns in the same communities, which caused intervention fatigue and confusion leading to disengagement in measles-rubella vaccination [27,49] and mass administration of ivermectin for onchocerciasis [60,61]. In Ethiopia integrating Vitamin A distribution into health services delivery, rather than community-based distribution campaigns, contributed to a decline in uptake [65].

In several studies a lack of information provided to volunteers prior to implementation or by community-based volunteers to community members led to reduced uptake of measles-rubella vaccinations [40], and ivermectin administration [32,66].

Several ivermectin distribution interventions reported the perceived need of treatment by the community had impacted uptake [32,60,67], with some community members citing preference for traditional medicines as more effective for onchocerciasis [63]. In immunization [26,40,58] and mass drug administration campaigns for onchocerciasis [32,33,37], the availability and training of community members (often as 'volunteers') in implementation was central to uptake. Community-buy-in was also facilitated through the involvement of religious groups, health committees and market associations which had legitimized the health messaging being delivered [60]. In schools, vaccination uptake increased when principals and teachers acted as advocates to engage parents [49].

Community trust was identified as a key barrier to successful implementation of interventions [27,32,40,60,61,67,68,71]. In several studies, distrust of the intervention was associated with fear of treatment (ivermectin and vaccinations). Experienced or believed adverse effects and treatment effectiveness, were reportedly leading to lower intervention uptake in Democratic Republic of Congo [33], Guinea [40], Tanzania [32], Ethiopia [64] and Cameroon [63,68]. Provision of free treatment, or provision of treatment by tribal members that differed to the community also caused suspicion. Several onchocerciasis interventions reported community drug distributors were not trusted by the community [68], due to perceived lack of professionalism [40] or lack of professional identification (e.g. a badge or certification) [61], lack of treatment knowledge [32], language barriers [60], and existing interpersonal conflicts [27,67]. In Cameroon community distributors reported being physically abused by community members [63].

## Surveillance, screening & detection

Twenty-six articles included engaging communities in vector surveillance, screening and detection of vision impairment or blindness using volunteers including key informants, school staff, community leaders and vector collectors (Table 5).

**Factors related to engagement of community in implementation.** Restricted availability and capacity of community members, including teachers and key informants, to conduct screening and detection of vision impairment or eye diseases posed barriers to implementation. For example, limited availability of teachers to conduct vision screenings in rural schools in Peru [20] or specialist teachers to provide training to parents of vision impaired children in rural Malawi [53]; created barriers to interventions targeting child eye health. In a school-based eye health intervention in India, time-poor teachers refused to participate in the

**Table 5. How community actors engaged in eye care surveillance, screening and detection.**

| Community actor(s) | Surveillance, screening and detection role(s) |
| --- | --- |
| Volunteers including key informants | • Detect childhood vision impairment [39,41,42,53,73–77]<br>• Conduct disease surveillance including measles [26]<br>• Defaulter tracing of missed vaccinations [28]<br>• Detect trachomatous trichiasis [44,78]<br>• Clear vegetation for vector control [79]<br>• Conduct surveillance of Simulian fly vectors [80]<br>• Detect individuals requiring eye care [81]<br>• Contribute to designing routine health and eye care [82] |
| Community and religious leaders | • Select volunteers to be trained in detecting childhood vision impairment [41] |
| School principals | • Select teacher to be trained in vision screening [24,83] |
| Teachers | • Provide vision screening to detect eye and vision problems in children [17,20,24,83–85]<br>• Coordinate vision screening [22] |
| Vector collectors | • Conduct surveillance of Simulian fly vectors [86] |

evaluation component of the intervention [24]. In Malawi, key informants were found to be more effective in detecting vision impairment or blindness compared to health surveillance assistances, however the door-to-door approach was reported to be time consuming [75]. Training teachers to screen children in India led to an increased sense of responsibility and knowledge of eye care [24]. However, when training was too basic or limited, teachers in Pakistan reported being confused about the processes [84]. Simplified training and use of resources (e.g. a screening card to detect trichiasis) facilitated implementation of a trachoma intervention in Tanzania [44]. Engaging stakeholders in pre-determined targets and initial intervention results assisted in developing leadership and adoption of school-based vision screening programs in Peru [20], and India [23]. In school-based programs, schools were more likely to participate in eye health interventions where endorsement letters from local government were provided [23].

Anganwadi workers in India [73] and key informants in Malawi [75] trained to detect individuals with vision impairment/blindness or need for eye care were motivated by their own desire to help their communities. In other contexts, non-financial incentives had facilitated involvement in screening and detection interventions including access to no cost eye examination for teachers [23], vision screening training among teachers leading to 'train the trainer' opportunities [24], and t-shirts and certificates for key informants in detecting children needing eye surgery in Tanzania [76]. Gender-based aspects of screening and detection included: localized knowledge of female community health volunteers in Nepal, and key informants in India, leading to easier identification of visually impaired children and provision of low vision aids [73,74]; understanding the need for gender segregated school-based vision screening in Pakistan [84]; and awareness of gender bias in Nepal, Uganda, and Malawi among female community health workers facilitated access to eye care for girls through advocacy and parent engagement [73].

In rural Peruvian communities where secondary or tertiary eye services took more than a day to reach, providing outreach child eye care services was more feasible [20]. However, some communities in India could not be reached for screening and surveillance [74].

## Referral & pathway navigation

Eighteen included articles involved strengthening patient referrals and navigation pathways through using community volunteers, children, teachers and technology and registries for reminders (Table 6).

**Factors related to engagement of community in implementation.** Anganwadi workers in India were reported to have strong relationships with hospitals, enabling referral of community members to cataract surgery [73] and played a key role in preparing lists of community members who were due for immunization [54]. Female community health volunteers in Nepal were central to referring or bringing in children with obvious eye problems to cataract

**Table 6. How community actors engage in strengthening eye care referrals and pathways.**

| Community actor(s) | Referral and pathway navigation role(s) |
|---|---|
| Teachers | • Ensure children referred reach relevant eye care [20,73,83]<br>• Provide referrals to children who fail vision screening [20,84] |
| Community health workers | • Prepare lists and recommend appointments for those requiring immunization [28,54]<br>• Refer community members requiring eye care [29,73,87] |
| Volunteers | • Refer children requiring eye care [29,39,75]<br>• Arrange transport for referred children requiring eye care [39]<br>• Refer cases of trachomatis trichiasis to surgeons [78] |
| Children | • Provide information to parents regarding own vision impairment |

outreach clinics [73] In Tanzania, properly educated community health workers who used a 'Frequently Asked Questions' resource with participants facilitated referral to trichiasis surgery; as a result participants who had received trichiasis surgery encouraged others to do so, citing benefits and short recovery time [29]. Fear of being treated poorly at hospital hampered referral to trichiasis surgery for patients in Egypt [31]. Health surveillance assistants in Malawi reported being too busy to conduct door-to-door vision screening, however it was noted that identification and eye health education were being delivered in immunization clinics [75]. In school-based programs, teachers lacked time or resources (e.g. guidelines) to facilitate referral of children to eye care [84]. Where rural children were referred, low uptake was associated with geographic distance and travel times [20]. Using children to engage parents in a refractive error referral intervention in India resulted in a low intervention response rate [88].

## Discussion

Our review identified studies that engaged community actors in the delivery of eye health interventions, and presented a synthesis of the barriers and facilitators to intervention implementation across a range of diverse and complex low- and middle-income contexts. The IPEC vision recognizes the importance of engaging communities in the delivery of eye health services and interventions, with the aim of providing a continuum of care across the eye health spectrum and life course [7]. However, as evidenced in our review, there remains significant gaps in community engagement both in the approaches taken, outcomes achieved, and the eye health areas targeted.

Using the WHO's community engagement framework [14] to assess 'levels' of participation (Table 2), our review revealed that the majority of identified eye care interventions at best only 'involved' community actors and members, and operated largely with external agency support. Such approaches limit the sustainability of intervention implementation, and longer-term capacity building, resulting in continued dependency on external, often high-income country based (S2 Table) support and funding. Interventions that were categorized as 'empowering' communities, were primarily onchocerciasis control programs employing 'Community Directed Treatment with Ivermectin' (CDTI) approaches that required consistent community engagement for a minimum of ten and up to twenty years. While intervention regularity and frequency varied across CDTI studies, longer timeframes supported sustained community engagement [33,37,60,64,66,68,71,89]. This finding underlines the need for community engagement approaches to actively shift towards empowerment and ownership [90] to ensure that integrated eye care is realized beyond the life of resource restrictive and short-term interventions.

Sustainability is particularly crucial in low resource settings, as limited funding has largely placed the role of community members in eye care interventions within the realm of volunteerism [27,33,34,37,57,61]. A lack of remuneration or valued incentives has led to low motivation and involvement [33], and decreased intervention uptake [60,63]. Similar findings from a rapid review [91] suggest that community engagement can have unintended negative consequences on involved individuals, including personal financial drain and fatigue. The issue stems partly from non-paid volunteerism being normalized in the delivery of eye care (and related) interventions in low- and middle-income settings, alongside the expectation that community members' goodwill is enough to propel engagement and motivation. This was corroborated in a couple of studies [63,71], largely because community volunteers were themselves in poverty afflicted contexts that did not allow time for unpaid labor. Conversely, adequate remuneration (e.g. Anganwadi workers in India) allowed sufficient human resources to be dedicated to the intervention resulting in increased community engagement [54].

To a greater degree, the sustainability issue stems from a broader macroeconomic environment in low- and middle-income settings, where even resources to support community health workers and other local level public health cadres are extremely limited, despite their cost-effectiveness [92]. Local and mid-level health workers are well-positioned to recruit and train community level workers or volunteers. Yet, as exemplified in various Sub-Saharan African contexts, poor availability of local and mid-level health workers, resulted in limited supervision, support and training provided to community volunteers [62,64,66,69]. These issues highlight the financing of community engagement activities, and the involvement and funding of local and mid-level health workers in the delivery of eye health interventions, as key barriers to the IPEC vision.

As identified in our review, non-health professionals can be effectively deployed to integrate eye care engagement programs in community settings (e.g. teachers for school eye care programs), however adequate training and resources, and simple protocols to support implementation are necessary. Trust was identified as a key principle and facilitator in community engagement, with enabling mechanisms including the development of advisory groups, and consulting and involving community leaders [33,37,66]. Yet the involvement alone of community members as deliverers of eye health interventions, does not guarantee community trust. This was particularly the case in CDTI programs, where fear of ivermectin use due to adverse side effects had made implementation difficult [32,33,63,64,68]. The legitimacy or professionalism of community volunteers was challenged in several interventions, highlighting the need to improve aspects and perceptions of credibility through quality of training, and visual signifiers such as uniforms and identification badges [32,40,60,61].

Our review revealed several gaps in the literature. First, despite the complexities of gender norms and biases in different cultural contexts being key to both uptake [84] and equity [73] of interventions, few studies targeted gender equity specifically despite known disparities in the burden of blindness and vision impairment for women and girls worldwide [1]. This finding supports the results of a systematic review that found limited evidence of interventions targeting gender equity in eye care [11]; suggesting that the role of community volunteers in circumventing gender inequity combined with targeted gender-focused intervention designs as potential solutions. Secondly, very few interventions targeting cataract surgery were identified, despite cataract being the main cause of blindness globally [93]. While cataract uptake has focused largely on surgery outreach, and reducing direct and indirect costs, evidence from a systematic review [10] reported the effectiveness of using successfully operated patients as champions and councilors, and suggested further research on familial connections and surgery uptake. Conversely, there was a high proportion of community engagement interventions targeting onchocerciasis, and primarily among children rather than adults. This may have been due to CDTI approaches, despite treatment uptake and elimination having varying success based on the level of participation from communities. Lastly, few trachoma interventions were identified compared to onchocerciasis. This may be due to antibiotic treatment as part of SAFE strategies not emphasizing community-directed treatment interventions to the same degree as onchocerciasis programs [94]; or due to a limitation in the community engagement terms used in our search strategy, which may not have aligned with the terminology used in SAFE interventions and programs.

## Limitations

This scoping review was limited to peer-review articles, and therefore may have missed potentially relevant information in grey literature articles, books and theses. Due to the heterogeneity of outcomes reported, we are unable to determine whether the role of community engagement has directly improved eye health outcomes.

## Conclusion

Overall, we present a variety of literature that contributes to 'bigger picture' approaches and research by identifying where community engagement would benefit and acknowledging gaps in low- and middle-income resource settings. When considering community engagement, structural, sociocultural, environmental, economic and agentic factors need to be considered to enable the full potential of empowering communities within an integrated and people-centered approach to eye care. Targeted supports for community health workers and volunteers, including financial reimbursements, incentives, and simple training resources may be required to sustain their impact and overcome implementation barriers to community engagement interventions across the spectrum of eye care needs.

## Supporting information

**S1 Checklist. Preferred Reporting Items for Systematic reviews and Meta-Analyses extension for Scoping Reviews (PRISMA-ScR) checklist.**
(DOCX)

**S1 Table. Search terms/strategy.**
(DOCX)

**S2 Table. Characteristics of included studies.** *Multiple = At least 3 funding sources associated with supporting the study are acknowledged; ^Epub ahead of print; CDD = community drug distributor; CDTI = community-directed treatment intervention; CHW = community health worker; FGD = focus group discussion.
(DOCX)

## Acknowledgments

We acknowledge Yeneneh Mulugeta Deneke, Sarity Dodson and Debbie Muirhead for providing feedback on the protocol and technical advice.

## Author Contributions

**Conceptualization:** Ling Lee, Elise Moo, Aryati Yashadhana.

**Data curation:** Ling Lee, Tiffany Angelopoulos, Aryati Yashadhana.

**Formal analysis:** Ling Lee, Tiffany Angelopoulos, Aryati Yashadhana.

**Funding acquisition:** Elise Moo.

**Methodology:** Ling Lee, Elise Moo, Aryati Yashadhana.

**Project administration:** Ling Lee, Elise Moo.

**Validation:** Tiffany Angelopoulos.

**Writing – original draft:** Ling Lee, Aryati Yashadhana.

**Writing – review & editing:** Ling Lee, Elise Moo, Tiffany Angelopoulos, Aryati Yashadhana.

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
