## [Decision Letter · Decision Letter 0]

1 Nov 2022

PONE-D-22-22189Integrated people-centered eye care: A scoping review on engaging communities in eye care in low- and middle-income settingsPLOS ONE Dear Dr. Lee,

Thank you for submitting your manuscript to PLOS ONE. After careful consideration, we feel that it has merit but does not fully meet PLOS ONE’s publication criteria as it currently stands. Therefore, we invite you to submit a revised version of the manuscript that addresses the points raised during the review process.

ACADEMIC EDITOR:

The reviewers have returned a decision and one of the reviewers has the following queries which I would like the authors to address:

Reviewer 1:

On Line 249: Did the author mean 'lacking promotion of onchocerciasis prevention and control beyond the life of the project"?

Line 267 to 268: Were the reasons for male distributors less likely to access female due to religious or cultural beliefs?

We look forward to receiving your revised manuscript.

Kind regards,

Godwin Ovenseri-Ogbomo, OD, MPH, PhD

Academic Editor

PLOS ONE

Journal Requirements:

Reviewers' comments:

Reviewer's Responses to Questions

**Comments to the Author**

1. Is the manuscript technically sound, and do the data support the conclusions?

Reviewer #1: Yes

Reviewer #2: Yes

2. Has the statistical analysis been performed appropriately and rigorously? 

Reviewer #1: N/A

Reviewer #2: Yes

3. Have the authors made all data underlying the findings in their manuscript fully available?

Reviewer #1: Yes

Reviewer #2: Yes

4. Is the manuscript presented in an intelligible fashion and written in standard English?

Reviewer #1: Yes

Reviewer #2: Yes

5. Review Comments to the Author

Reviewer #1: The article is well planned and the rationale for the review is presented. The authors report the processes and procedures undertaken—as well as any limitations of the approach.

On Line 249: Did the author mean 'lacking promotion of onchocerciasis prevention and control beyond the life of the project"?

Line 267 to 268: Were the reasons for male distributors less likely to access female due to religious or cultural beliefs?

Good review that has identified the gaps in community engagement relating to IPEC.

Reviewer #2: The review is well conducted and elaborate it provides evidence on the breadth of eye care interventions and the factors to be considered when engaging and empowering communities in integrated people-centered eye care programs. This is useful in the consideration for the achievement of universal health coverage. A paper well written.

6. PLOS authors have the option to publish the peer review history of their article (what does this mean?). If published, this will include your full peer review and any attached files.

Reviewer #1: **Yes: **Tuwani A. Rasengane

Reviewer #2: No

---

## [Author Response · Author response to Decision Letter 0]

23 Nov 2022

Reviewer 1:

On Line 249: Did the author mean 'lacking promotion of onchocerciasis prevention and control beyond the life of the project"?

Response: Thank you for identifying our error. We have revised the manuscript to include ‘prevention and control’

Line 267 to 268: Were the reasons for male distributors less likely to access female due to religious or cultural beliefs?

Response: Male distributors were less likely to access females due to cultural and health seeking behavioural norms. Additional details have been included.

Journal Requirements

Response: No retracted papers have been cited in the manuscript. Where required for consistency, journal abbreviations, doi links have been edited. Please get in contact if further amendments are required.

---

## [Editor Report · Decision Letter 1]

25 Nov 2022

Integrated people-centered eye care: A scoping review on engaging communities in eye care in low- and middle-income settings

PONE-D-22-22189R1

Dear Dr. Lee,

We’re pleased to inform you that your manuscript has been judged scientifically suitable for publication and will be formally accepted for publication once it meets all outstanding technical requirements.

Kind regards,

Godwin Ovenseri-Ogbomo, OD, MPH, PhD

Academic Editor

PLOS ONE

Additional Editor Comments (optional):

Thank you for revising the submitted manuscript.
---

## [Editor Report · Acceptance letter]

9 Jan 2023

PONE-D-22-22189R1 

Integrated people-centered eye care: A scoping review on engaging communities in eye care in low- and middle-income settings 

Dear Dr. Lee:

I'm pleased to inform you that your manuscript has been deemed suitable for publication in PLOS ONE. Congratulations! Your manuscript is now with our production department. 

Kind regards, 

on behalf of

Dr. Godwin Ovenseri-Ogbomo 

Academic Editor

PLOS ONE